



**Assessment of Landslide Susceptibility using Weight of Evidence and Frequency**
**Ratio Model in Shahpur Valley, Eastern Hindu Kush**
[1]*Ghani Rahman,[2] Atta Ur Rahman, [3]Alam Sher Bacha, [4]Shakeel Mahmood,
[5]Muhammad Farhan Ul Moazzam, [5]*Byung Gul Lee
[1]Department of Geography, University of Gujrat, Pakistan
[2]Department of Geography, University of Peshawar, Pakistan
[3]National Center of Excellence in Geology, University of Peshawar, Pakistan
[4]Government College University, Lahore, Pakistan
[5]Department of Civil Engineering, College of Ocean Sciences, Jeju National University, South Korea
Corresponding author: ghanigeo@gmail.com  leebg@jejunu.ac.kr
**Abstract**
This study assessing the landslide susceptibility using Weight of Evidence (WoE) and
Frequency Ratio (FR) model in Shahpur valley, situated in the eastern Hindu Kush. Here, landslide is
a recurrent phenomenon that disrupts natural environment and cause huge property damages as well
as incurs human losses every year. These damages are expected to increase due to high rate of
deforestation in the region, population growth, agricultural expansion and infrastructural
development on the fragile slopes. Initially, landslide inventory map was prepared from SPOT5
satellite image and were verified from frequent visits in the field. Seven landslide contributing factors
including surface geology, fault lines, slope aspect and gradient, land use, proximity to roads and
stream were selected. To analyze the relationship of landslide occurrence with these causative
factors, WoE and FR models were used. Based on WoE and FR model landslide susceptibility
zonation maps were prepared and were reclassified into very low to very high landslide susceptible
zones. Finally, the resultant maps of landslide susceptibility were authenticated using success rate
curve and prediction rate curve approach to validate the models.
**Keywords:** Landslide Susceptibility, Weight of Evidence, Frequency Ratio, Success rate curve,
Prediction rate curve
**1. Introduction**
Globally, the frequency of geological and hydro-meteorological disasters is increased in the
last two decades with devastating consequences (Rahman et al. 2017). Landslide is among the
geological hazards that cause damages to human life, their property and infrastructure (Jehan &
Ahmad 2006). The Hindu Kush-Himalayan (HKH) is young mountain system where landslides,
avalanches, floods and earthquakes are very common (A. Rahman & Shaw, 2014; G. Rahman,
Rahman, Samiullah, et al., 2017). In this region landsliding is a recurrent phenomenon and mostly
been initiated by seismic activity or prolong rainfall (Kamp et al., 2010a; Regmi et al., 2014; G.
Rahman, Rahman, & Collins, 2017). The frequent landslide events have been causing damages to



property, infrastructure and sometimes led to human losses. Kanungo et al. (2009) repoeted that the global share of landslides was five percent among all the natural hazards during 1990-2005and tend to increase in future because of seismic activities, increasing rainfall intensities and anthropogenic activities on the fragile slopes(Pareek et al., 2010; Conforti et al., 2014; G. Rahman, Rahman, & Collins, 2017).

Landsliding is one of the complex geomorphic process (Nandi & Shakoor, 2010; Allen et al., 2011) mainly triggered by area geology, seismicity, drainage pattern, land cover, gradient and rainfall (Sudmeier-Rieux et al., 2012; G. Rahman, Rahman, Samiullah, et al., 2017). The occurrence of landslides has significant relationship with the slope gradient, aspect, vegetation cover and soil thickness of the slope(Sengupta et al., 2010); Rahman et al. 2011). Prolong rainfall in mountainous areas with fragile slope also increases probabilities of the landslide occurrence. The seismic activities and lithology are other important factors affecting the slope stability (C. Van Westen et al., 2010; A. Rahman et al., 2011). Similarly anthropogenic activities in terms of road construction, expansion of human settlement, deforestation and expansion of agricultural activities on fragile slope further intensifies the landslide susceptibility (Rahman et al. 2017).

The landslides occur throughout the world particularly in certain hotspots (Nadim et al., 2006). Many studies have been conducted to explore the impacts of landslides on human lives, property and infrastructure. A diminutive attention has been given to landslide impacts on the natural environment (Schuster & Highland, 2007). Similarly, attention has been paid to the role of landslides in disturbance of ecological system. The environmental effects caused by landslides are changes in agricultural activities, changes to natural ecosystems, changes in river morphology because of landslide dams (Nakamura et al., 2000). Other effects included sedimentation in river channels and flash flood due to breaching of landslide dams. Landslides also disturbs the natural habitat of certain endanger species in susceptible zone. The landslide events also effects biodiversity of the affected area, therefore strict forest preservation measures are highly required to reduce environmental damage (Geertsema & Pojar, 2007).

Landslide susceptibility is basically the geo-spatial probability of slope failure. The landslides occurrence depends on the presence of some geo-environmental factors(Guzzetti et al., 2005). During past decade, numerous scientific studies including Lee,(2004), Chen and Wang,(2007), Kavzoglu et al.,(2014), Bourenane et al.,(2016), Ding et al.,(2017) and G. Rahman, Rahman, Samiullah, et al.,(2017)have been conducted regarding the fragile mountains and established a wide range of empirical approaches for analyzing landslide susceptibility to identify the extent of potentially susceptible landslide areas. Quantitative, semi-quantitative and qualitative techniques including



statistical and deterministic approaches has been used in various studies to assess landslide
susceptibility or hazard zones(C. J. Van Westen et al., 2008). The landslide indices use the semi-
quantitative, quantitative and qualitative methods for identification of areas having similar
characteristics with respect to geological and geomorphological settings of the landslide prone areas
(Kouli et al., 2010). Qualitative methodologies use rating procedure, indigenous knowledge and
weighting procedures forming bases for semi-quantitative methods. However, quantitative methods
used statistical techniques to find out the relationship between causal factors and landslide
events(Ayalew & Yamagishi, 2005).

The spatial probability of landslides can be predicted by applying various quantitative
methodologies like frequency ratio, information value, weight of evidence, fuzzy neural network,
logistic regression and many others. These methods depend on inventory of past landslides and
thematic maps of landslide causative factors(Hussin et al., 2016). In recent years, geospatial
technology is widely applied in studies regarding landslide susceptibility mapping, risk identification
and management (Akbar & Ha, 2011). Geospatial technology provides a framework for mapping the
past landslide events and combine the landslide causative factors for producing landslide
susceptibility map and therefore it has become an integral part of landslide susceptibility zonation
(LSZ).

The HKH is an active seismic region and hence most of the landslides have also been initiated
by seismic activity (Kamp et al., 2010b). Developmental work is usually affected by the frequently
occurring phenomena of landsliding in the HKH region. It is therefore, a dire need of time to identify
the landslide prone areas that will not only minimize the risk of landsliding in future but will also
provide base for the future planning as well. In present study the landslide susceptibility mapping is
based on frequency ratio and weight of evidence model to develop landslide susceptibility maps of
Shahpur valley, HKH region.
**2. The Study Area**

The study area, Shahpur valley lies in the Hindu Raj Mountains. These mountains are
considered as the offshoot of Hindu Kush mountain system (Dichter, 1967). Moving from north to
south the height of these Hindu Kush Mountains tends to decreases. The latitudinal extent of the
valley is 34° 52′ 31″ to 35° 9′ 35″ while longitudinal extent is 72° 40′ 10″ to 72° 48′ 44″ as shown in
the Figure 1. The total area of Shahpur valley is approximately 259 square kilometers. Climatically,
Shahpur valley is the part of moist temperate zone. The valley receives heavy rainfall during summer
season from monsoon, while in winter at higher altitudes mostly precipitation occurs in the form of



heavy snowfall. Climate of the valley remain mild to warm in summer while temperature decrease to
chill cold in winter season throughout the valley (G. Rahman et al., 2019).

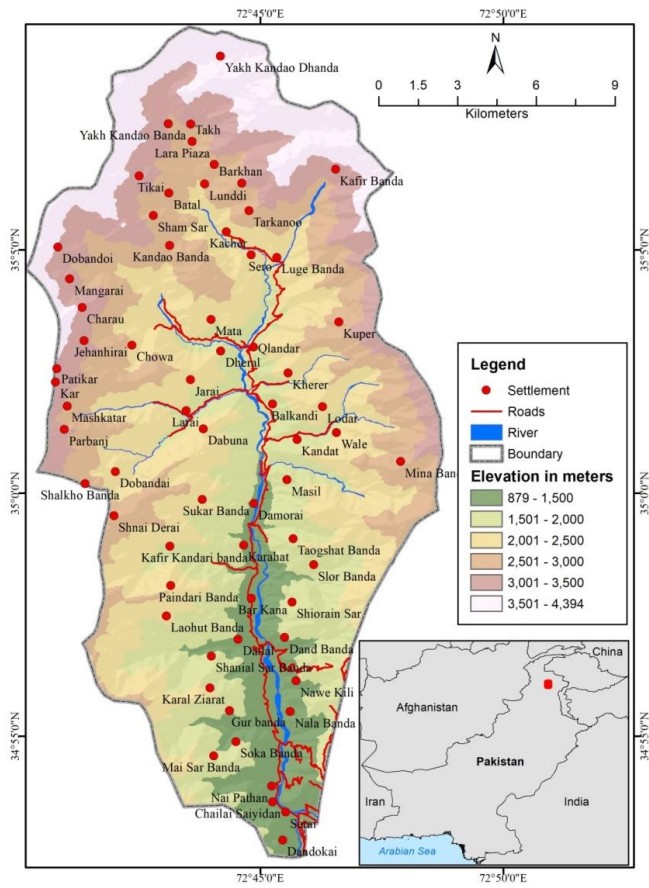

Figure 1: Digital Elevation Map of Shahpur valley

HKH region came into existence due to the collision of Eurasian and Indian plate during the

Cretaceous and Mio-Pliocene epoch. As a result of this collision these mountains are still
continuously rising   at a rate of 4 to 5 mm/year (Jehan & Ahmad, 2006). There is high altitudinal
variation of 3600 meters in just 259 square km area (Figure 1). The valley has steep slope in the
upper part while it became gentle in the lower reach of the valley. The valley is drained by a stream
known as Khan Khwar. The study area consist of young mountain system that have immature
geology and is prone to landsliding phenomena which often results considerable property damages
and human losses almost every. The probability of these damages is expected to increase further as a
result of anthropogenic activities like deforestation, overgrazing, agricultural activities and



development of infrastructure in this area. Population growth has posed more pressure on the fragile
slopes and has made it more vulnerable for landsliding.

**3. Methods and Material**
In the eastern Hindu Kush region, Shahpur valley was selected for detailed analysis to grasp
the governing landslide causative factors, which frequently trigger landsliding. The data from both
primary and secondary data sources were used to achieve the objectives of the study (Figure 2). The
past landslide sites were identified and mapped on 2.5m resolution SPOT image of April 2013. A
thorough field study was conducted to confirm the landslide sites on the ground and identify the
landslide triggering factors with local community knowledge. Seven triggering factors namely
surface geology, proximity to fault line, slope gradient and aspect, land use/ land cover, nearness to
road and streams were identified.
Data regarding landslide triggering factors were acquired including the surface geology and
tectonics from geological map of North Pakistan. The administrative boundaries and settlement
shape-files was prepared from topographic sheets (RF 1:50,000) obtained from survey of Pakistan.
Spatial features of roads network was acquired from the office of Communication and Works
Department, Peshawar. Land use/land cover map was obtained after applying supervised
classification on SPOT satellite image using ArcGIS 10.2. ASTERGDEM having 30m was used for
extracting slope angle, slope aspect and hydrology of the study area. Furthermore, a detailed field
survey was conducted to validate the sites of already activated and potentially active landslide area.
GIS and Remote Sensing have been used for the preparation of spatial databases and
landslides inventory map. Weight of evidence and frequency ratio model analysis is a bivariate
statistical methodology in which the importance of each factor or combined factors is individually
analyzed with respect to spatial distribution of existing landslides. The assumption in both models is
that the factors which influenced the incidence of landslides in the past will be the same to trigger
new landslides in future.

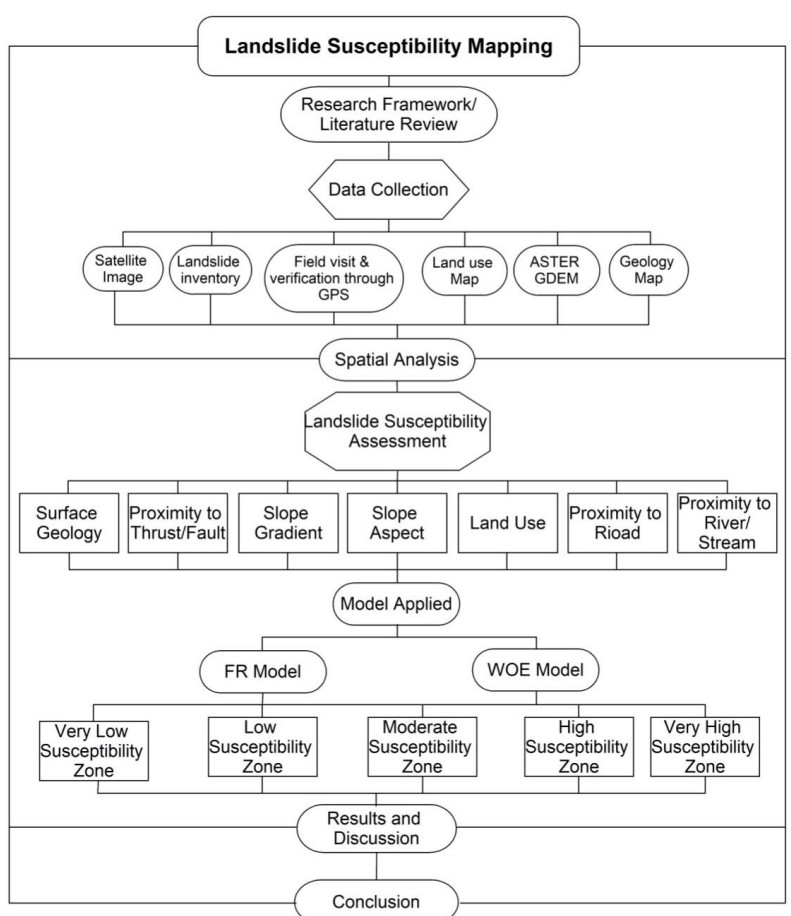

Figure 2: Research Model

## 3.1 Weight of Evidence Model

Weight of evidence model (Bonham-Carter et al., 1989; Bonham-Carter, 1994) is based on

Eq. 1 and Eq. 2:

$$W^+ = \ln \frac{P\left(\frac{B}{D}\right)}{P\left(\frac{B}{\bar{D}}\right)} \qquad (1)$$

$$W^- = \ln \frac{P\left(\frac{\bar{B}}{D}\right)}{P\left(\frac{\bar{B}}{\bar{D}}\right)} \qquad (2)$$

In the above equations, $P$ is the probability while ln is the natural log. $B$ and $\bar{B}$ respectively

represent the presence and absence of potential landslide evidence factor. Likewise, $D$ and $\bar{D}$  is the
presence and absence of landslide respectively. For the calculation of weight of each causative





factors contributing in landslide occurrence Eq.3 and Eq.4 have been used after (C. Van Westen et
al., 2003).
$$W^+ = \ln\left\{\left(\frac{[Npix1]}{[Npix1]+[Npix2]}\right) / \left(\frac{[Npix3]}{[Npix3]+[Npix4]}\right)\right\} \qquad \text{(Eq.3)}$$
$$W^- = \ln\left\{\left(\frac{[Npix3]}{[Npix1]+[Npix2]}\right) / \left(\frac{[Npix4]}{[Npix3]+[Npix4]}\right)\right\} \qquad \text{(Eq.4)}$$
Where the $Npix1$ is the number of pixels express the existence of both landslide contributing
factor and landslides; $Npix2$ represent the presence of landslide and absence of landslide contributing
factor. While $Npix3$ represent the presence of landslide contributing factor and absence of landslide.
Similarly, $Npix4$ represent the absence of both landslide and landslide contributing factors. Final
weight expressed with $W^c$ was calculated using Eq.5:
$$W^c = (W^+) - (W^-) \qquad \text{(Eq.5)}$$
Where, $W^c$ is the difference of $W^+$ and $W^-$. This elucidates the spatial relationship of all
landslide contributing factors and landslide.
**3.2 Frequency Ratio Model**
To analyze the effect of landslide contributing factors on the occurrence of landsliding was also
examined through frequency ratio model. It is a ratio of landslides occurred area with respect to the
total study area, and is also the proportion of the landslide occurrence probabilities to a non-
occurrence  for a given attribute (Bonham-Carter, 1994; Lee & Talib, 2005). In frequency ratio
model, a statistical value for each class of a factor map using the Eq.6:
$$FR = \frac{N_{pix(Si)}/N_{pix(Ni)}}{\sum N_{pix(Si)}/\sum N_{pix(Ni)}} \qquad \text{(Eq.6)}$$
Where, $N_{pix(Si)}$ is the number of landslide pixels containing class $i$, $N_{pix(Ni)}$ is the total number of
pixels of class $i$, $\sum N_{pix(Si)}$ is total number of landslide pixels in the entire study area, whereas
$\sum N_{pix(Ni)}$ is the total number of pixels of the entire study area.
**3.3 Landslide Susceptibility Index (LSI)**
LSI for both, frequency ratio and weight of evidence model was generated by combining the landslide
causative/ contributing factors in GIS based on the $W^c$ and $FR$ values for overlay analysis using the
Eq.7:
$$LSI = \sum W^c \, , LSI = \sum FR \qquad \text{(Eq.7)}$$


Where $\sum W^c$ is the total derived weight of weight of evidence model and $\sum FR$ is the total derived
weight of frequency ratio model.

**4. Results and Discussion**

In this paper frequency ratio and weight of evidence models are used with aim to determine
and geo-visualize the landslide susceptibility with resultant map is susceptibility zonation that has
been extensively applied in many parts of the world for landslides risk reduction(Shahabi et al.,
2015).

**4.1 Inventory of Landslides in Shahpur Valley**

The past landslides sites were marked on multi-spectral SPOT satellite image of April 2013. These
sites were verified in through series of field visits. About three hundred landslides of varying sizes
were marked on the satellite image and verified from field investigation in the study area (G. Rahman
et al., 2019) (Figure 3). This landslide inventory was randomly divided into two groups, group one
was taken as training landslides (80%) and the second group was taken as validation landslides
(20%). These landslides were then rasterized to find out the number of pixels in every class of a
factor map for calculation of frequency ratio and weight of evidence model values.

**4.2 Landslide Contributing/ causative factors**

Landsliding is a natural phenomenon and its occurrence is determined by variety of causative factors.
In this study, surface lithology/geology, stream buffer for assessing impacts of stream proximity, land
cover, slope aspect, slope gradient, fault line impacts and impacts of road network were selected as
landslides contributing factors (Figure 4). WoE and FRM statistical models based on correlation of
past landslide and causative factors were used to define the weight of each class of every factor map.
In WoE model the positive weight ($W^+$), negative weight ($W^-$) and contrast weight ($W^c$) while for
FR model the frequency ratio were calculated for each class of a contributing factor map (Table 1).

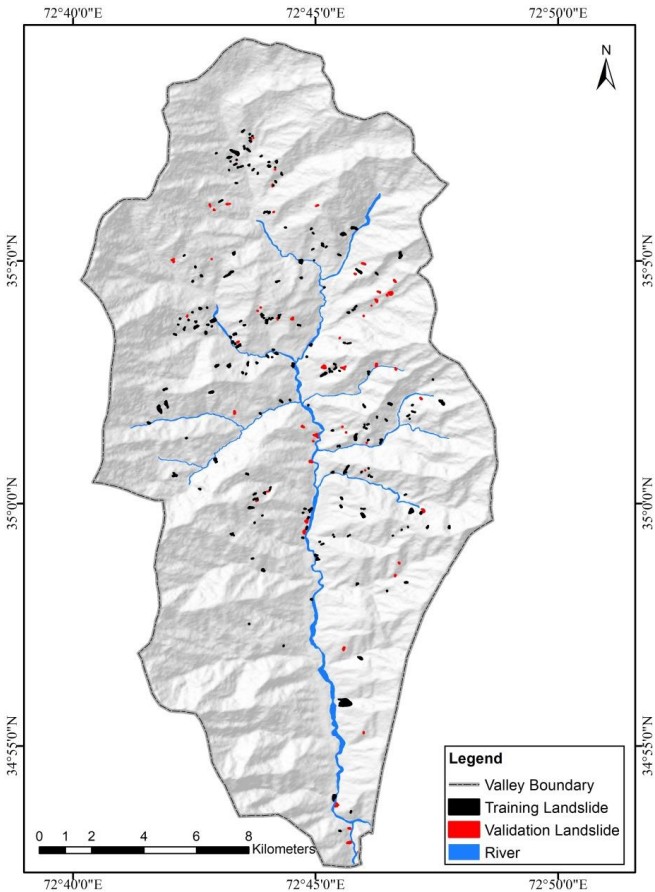

Figure 3: Shahpur valley, Landslide inventory and distribution of past landslides
**4.2.1 Surface Geology**

To assess the relationship of surface geology and landslide occurrence in Shahpur valley,

surface geology was taken as a causative factor and its relationship were assessed using WoE and
FRM. Surface geology types are shown in Figure 4g. The highest positive $W^c$ weight was found in
Darwaza Sar Potassic Granite Gneiss (0.71) and Alluvium (0.59). These both classes have very
positive correlation with landslides using WoE model. Alluvium in this region is of quaternary period
and is brought by Indus river and its tributaries derived from the Kohistan island arc terrane (Baig,
1990). Similar results were found in FR values. The highest negative correlation was in geology class
Jijal Ultramafics having $W^c$ value -3.64 and FR 0.03 (Table 1).
**4.2.2 Fault Line**

The occurrence of landslides has a strong correlation with fault lines (Korup, 2004; G.

Rahman, Rahman, & Collins, 2017). Fault lines existence at high slope gradient provides favorable





settings for slope failure. There is a complex tectonic structure in the study area and is considered as
causal factor in slope instability. It is evident form the analysis that the tectonic structures have strong
correlation with landslide occurrence. The highest positive $W^c$ value (1.56) was found in the area of
buffer zone 0-250 meters followed by 251-500 meters buffer zone and the lowest $W^c$ was in area of
greater than 1000 meters according to WoE model. Similar results was found in frequency ratio
model, the highest FR value (2.87) was in the buffer zone of 0-250 meters and the lowest was in area
of greater than 1000 meters area.
**Table 1.** Shahpur valley, calculated weight of each class of causative factors

| Classes | $N_{\text{pix (Si)}}$ | %age of $N_{\text{pix (Si)}}$ | $N_{\text{pix (Ni)}}$ | %age of $N_{\text{pix (Ni)}}$ | $W^+$ | $W^-$ | $W^c$ | FR |
|---|---|---|---|---|---|---|---|---|
| **Surface Geology** | | | | | | | | |
| Alluvium | 1499 | 18.52 | 290137 | 11.20 | 0.51 | -0.09 | 0.59 | 1.65 |
| Greenschist Melange | 806 | 9.96 | 165892 | 6.40 | 0.44 | -0.04 | 0.48 | 1.56 |
| Jabrai Granite Gneiss | 903 | 11.16 | 497979 | 19.22 | -0.54 | 0.10 | -0.64 | 0.58 |
| Alpuraicalc-mica-garnet schist | 990 | 12.23 | 235014 | 9.07 | 0.30 | -0.04 | 0.34 | 1.35 |
| Karora Group | 967 | 11.95 | 501955 | 19.37 | -0.48 | 0.09 | -0.57 | 0.62 |
| Besham Group | 1436 | 17.74 | 441986 | 17.06 | 0.04 | -0.01 | 0.05 | 1.04 |
| Manglaur Formation | 1218 | 15.05 | 378895 | 14.62 | 0.03 | -0.01 | 0.03 | 1.03 |
| Darwaza Sar Potassic Granite Gneiss | 271 | 3.35 | 43693 | 1.69 | 0.69 | -0.02 | 0.71 | 1.99 |
| Jijal Ultramafics | 3 | 0.04 | 35939 | 1.39 | -3.63 | 0.01 | -3.64 | 0.03 |
| **Fault Line Buffer (m)** | | | | | | | | |
| 0 – 250 | 4018 | 49.65 | 448304 | 17.30 | 1.06 | -0.50 | 1.56 | 2.87 |
| 251 – 500 | 2325 | 28.73 | 409420 | 15.80 | 0.60 | -0.17 | 0.77 | 1.82 |
| 501 – 1000 | 760 | 9.39 | 676634 | 26.11 | -1.02 | 0.20 | -1.23 | 0.36 |
| > 1000 | 990 | 12.23 | 1057133 | 40.79 | -1.21 | 0.40 | -1.60 | 0.30 |
| **Slope Gradient** | | | | | | | | |
| $0\text{-}5^0$ | 91 | 1.12 | 67722 | 2.61 | -0.85 | 0.02 | -0.86 | 0.43 |
| $6\text{-}15^0$ | 514 | 6.35 | 261492 | 10.09 | -0.46 | 0.04 | -0.50 | 0.63 |
| $16\text{-}30^0$ | 2138 | 26.42 | 668931 | 25.81 | 0.02 | -0.01 | 0.03 | 1.02 |
| $31\text{-}45^0$ | 4847 | 59.89 | 1366442 | 52.73 | 0.13 | -0.16 | 0.29 | 1.14 |
| $> 46^0$ | 503 | 6.22 | 226903 | 8.76 | -0.34 | 0.03 | -0.37 | 0.71 |
| **Slope Aspect** | | | | | | | | |
| Flat | 1 | 0.01 | 1004 | 0.04 | -1.14 | 0.00 | -1.14 | 0.32 |
| North | 503 | 6.22 | 214667 | 8.28 | -0.29 | 0.02 | -0.31 | 0.75 |
| Northeast | 531 | 6.56 | 284530 | 10.98 | -0.52 | 0.05 | -0.56 | 0.60 |
| East | 1444 | 17.84 | 387999 | 14.97 | 0.18 | -0.03 | 0.21 | 1.19 |
| Southeast | 881 | 10.89 | 395492 | 15.26 | -0.34 | 0.05 | -0.39 | 0.71 |
| South | 1775 | 21.93 | 366954 | 14.16 | 0.44 | -0.10 | 0.53 | 1.55 |
| Southwest | 1135 | 14.02 | 356943 | 13.77 | 0.02 | 0.00 | 0.02 | 1.02 |
| West | 819 | 10.12 | 317383 | 12.25 | -0.19 | 0.02 | -0.22 | 0.83 |
| Northwest | 1004 | 12.41 | 266520 | 10.28 | 0.19 | -0.02 | 0.21 | 1.21 |



| Land Cover | | | | | | | | |
|---|---|---|---|---|---|---|---|---|
| Range Land | 2762 | 34.13 | 847632 | 32.71 | 0.04 | -0.02 | 0.06 | 1.04 |
| Forest | 2621 | 32.39 | 1036194 | 39.98 | -0.21 | 0.12 | -0.33 | 0.81 |
| Glacier and Snow | 108 | 1.33 | 111086 | 4.29 | -1.17 | 0.03 | -1.20 | 0.31 |
| Agriculture Land | 2100 | 25.95 | 416925 | 16.09 | 0.48 | -0.13 | 0.61 | 1.61 |
| Settlement | 48 | 0.59 | 37521 | 1.45 | -0.89 | 0.01 | -0.90 | 0.41 |
| Barren Land | 87 | 1.08 | 87880 | 3.39 | -1.15 | 0.02 | -1.17 | 0.32 |
| Stream/torrent | 367 | 4.53 | 54252 | 2.09 | 0.78 | -0.03 | 0.80 | 2.17 |
| **Road Buffer (m)** | | | | | | | | |
| 0-100 | 769 | 9.50 | 130869 | 5.05 | 0.63 | -0.05 | 0.68 | 1.88 |
| 101-200 | 541 | 6.68 | 103117 | 3.98 | 0.52 | -0.03 | 0.55 | 1.68 |
| 201-300 | 591 | 7.30 | 92441 | 3.57 | 0.72 | -0.04 | 0.76 | 2.05 |
| 301-400 | 141 | 1.74 | 85731 | 3.31 | -0.64 | 0.02 | -0.66 | 0.53 |
| > 400 | 6051 | 74.77 | 2179333 | 84.10 | -0.12 | 0.46 | -0.58 | 0.89 |
| **Stream Buffer (m)** | | | | | | | | |
| 0-100 | 1918 | 23.70 | 294902 | 11.38 | 0.74 | -0.15 | 0.89 | 2.08 |
| 101-200 | 1555 | 19.21 | 265711 | 10.25 | 0.63 | -0.11 | 0.74 | 1.87 |
| 201-300 | 1021 | 12.62 | 255277 | 9.85 | 0.25 | -0.03 | 0.28 | 1.28 |
| 301-400 | 799 | 9.87 | 247979 | 9.57 | 0.03 | 0.00 | 0.03 | 1.03 |
| 401-500 | 395 | 4.88 | 238952 | 9.22 | -0.64 | 0.05 | -0.68 | 0.53 |
| >500 | 2405 | 29.72 | 1288669 | 49.73 | -0.52 | 0.34 | -0.85 | 0.60 |


### 4.2.3 Slope Gradient

Slope gradient affects the population distribution, their activities and distribution of natural
resources. Likewise, landslide distribution also has a close association with slope gradient and act as
a controlling factor in slope failure. Slope gradient has direct relation with slope failure and the
chances of landslide incidence escalate with increase in slope gradient. It was observed during field
visits that the high landslide density areas were on the slope along the road and stream where lateral
cutting was dominant factor. Map of the slope gradient for the study area was generated from
AsterGDEM having 30 meters spatial resolution in GIS (Figure 4c). The analysis of both WoE and
FRM shows that the role of 31-45 degree slope is higher in slope failure as the highest $W^c$ value
(0.29) and FR value (1.14) was found in this class of slope gradient (Table 1). While the slope
gradient 0-5 and 6-15 degree class has negative correlation with landslide.

### 4.2.4 Slope Aspect

Slope aspect does not have a direct impact on landslide occurrence, but indirectly accelerate
the landslide process. The sunlight intensity and duration, amount of rainfall and moisture holding
capacity and distribution of vegetation all are affected by slope direction. The analysis reveals that
the south facing slope has very strong positive correlation with landslide as the value of $W^c$(0.53)


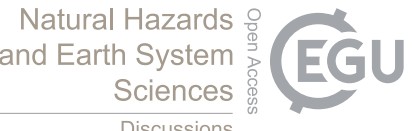

and FR (1.55) is higher in this class followed by northwest $W^c$(0.21) and FR (1.21) facing slope
(Table 1). In the study area, high landslides in south facing slopes may be due to its high exposition
to sunlight and receiving ample amount of rainfall as of windward side.


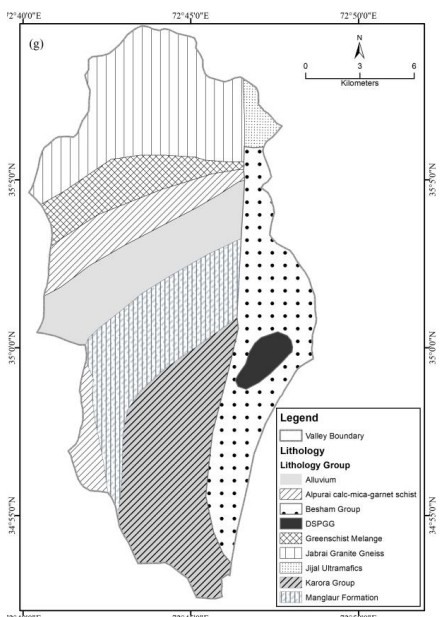


**Fig. 4.** Shahpur valley: (a) Land use map; (b) Slope aspect; (c) Slope gradient; (d) proximity to road;
(e) Proximity to stream; (f) Proximity to fault lines; (g) Surface geology


**4.2.5 Land Use/ Land Cover**

The forest cover protect the mountainous slope from weathering and mass wasting processes
as the roots hold the underneath soil and keep the slope stable. Increasing population growth has
increase the demand of wood and land for food has disturbed the slope of almost all the mountainous
region of the world and have led to slope instability. Land cover of Shahpur valley was developed
from the SPOT satellite of image (Figure 4a). Analyzing the influence of land use/ land cover on
landslide, statistical weight for each class of the land use was calculated using WoE and frequency
ratio model. The highest weight of both WoE ($W^c = 0.80$) and FR (2.17) was found for
stream/torrent class. This was because in the study area the stream/torrent has high lateral erosion and
thus initiates new slides. The second high positive correlation was of agriculture land with landslide.
In the study area forest cover are mostly cleared for agriculture activities. Agriculture practice is on
terrace field which also make the slope susceptible to landslide. It was found from the analysis that
barren land has negative correlation with landslide as in the study area the land was barren because of
presence of hard rock masses which does not support any vegetation in the higher slopes.




**4.2.6 Proximity to Road**
The road constructions often disturb the slope and expedite the weathering and mass wasting process
thus increase the probability of landslide occurrence. It also provides means of accessibility and
accelerates the process of deforestation. In the current study, proximity to road is used as a causative
factor of landslide. The results show high positive correlation with road proximity up to 300 meter.
The highest $W^c$ value (0.68) and FR (1.88) was found in 0-100 meters road proximity. This elucidate
that the slope near to road have more probability to slope failure.
**4.2.7 Proximity to Stream/torrent**
In order to examine the relationship of stream/torrent on landslide, WoE and frequency ratio
statistical models were applied. It was found from the analysis that both WoE and FRM have higher
value near the stream that indicates high probability in this region. The highest $W^c$ (0.89) and FR
value (2.08) were found in the proximity of 0-100 meters (Table 1). The results show that the region
up to 400 meters of proximity to stream shows the positive correlation toward the landslide
probability. The highest negative correlation was found in the region of greater than greater than 500
meters of stream.
**4.3 Landslide Susceptibility Zonation**
Landslide is the common menace to the property, human lives and infrastructure in Shahpur
valley. For its mitigation the first utmost important step is to identify high susceptible landslide areas.
LSZ map divide the region into very low to very high susceptible zone according to their
susceptibility based on integration of landslide causal factors. GIS provides framework for
integration of different landslide causal factors to produce LSZ map. To minimize subjectivity,
quantitative weight to each class of factor maps was applied based WoE and FR models for
generation of LSZ map of Shahpur valley. The LSZ map was created based on both WoE and FR
models by summing all the relative weight of each class of factor maps using following expressions:
$$LSI = \sum W^c \qquad (8)$$
$$LSI = \sum FR \qquad (9)$$
Where $\sum W^c$ isthe total derived weight of each class of the factor maps for WoE model, while
$\sum FR$ is the sum of the derived weight of each class of the factor map of frequency ratio model. In
both cases the higher the value of LSI, greater would be the probability of landslides incident. Based
on LSI, the study area was divided into zones of Very high to very low Susceptibility.

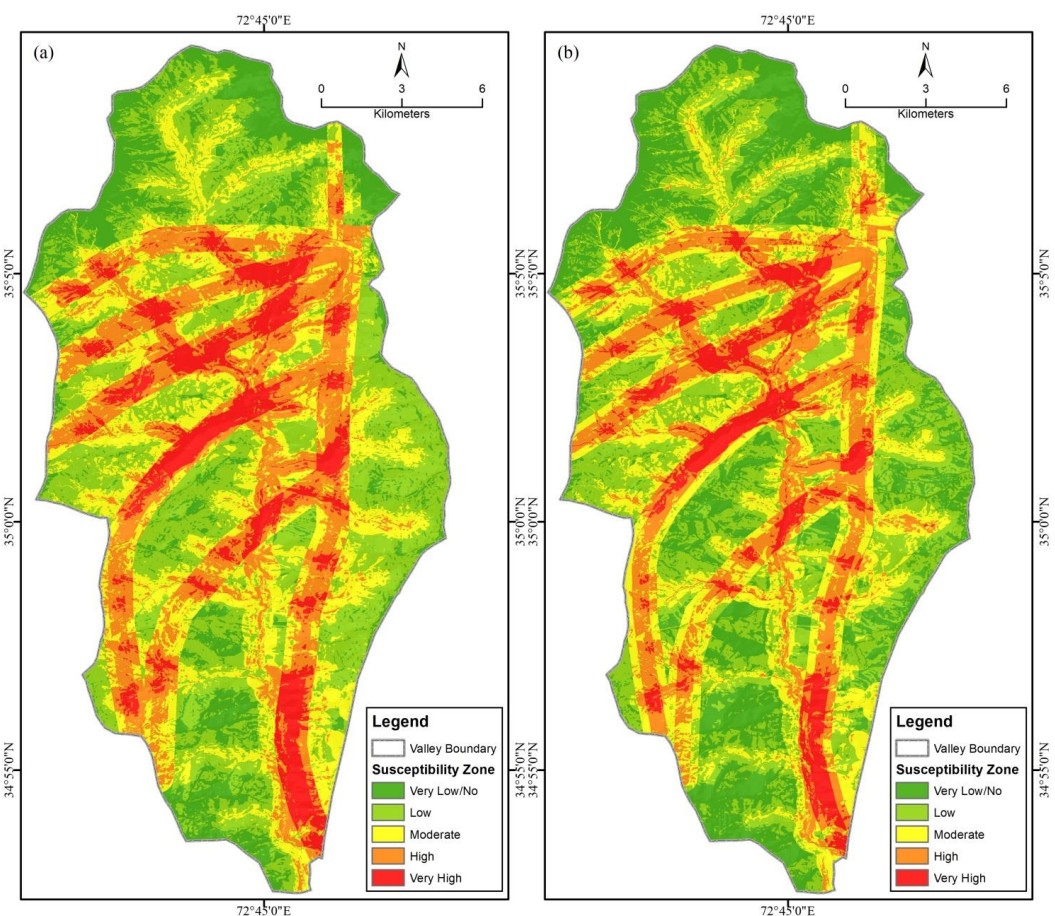

**Fig. 5.** Shahpur valley, (a) landslide susceptibility zones based on WoE; (b) landslide susceptibility zones based on FR

## 4.4 Validation of Landslide Susceptibility Map

The landslide susceptibility map was validated using success rate curve based on training landslide that were 80% of the total landslide inventory and prediction rate curve using validation landslides that were 20% of the total landslide inventory. The success rate curve and prediction rate curve elucidates the accuracy of WoE and FRM for selected causative factors to landslide occurrences. Success rate curve and prediction rate curve was calculated using the LSI values ranging from highly susceptible to very low susceptible class and overlaid with the existing layer of landslide area through geo-statistical tool in GIS. Cumulative percentages for both susceptibility class and landslide area were calculated and susceptibility class was plot on x-axis and landslide area on y-axis to generate both success rate curve and prediction rate curve. Both success rate curve and prediction



rate curve have steep curve which indicates significant result for both WoE and FR models. Both the
susceptibility maps prepared based on WoE and FR models were validated using area under (AUC)
technique. It is a quantitative measurement of success rate and predictive rates of the landslide
susceptibility map. The AUC for WoE model was 87.92% for success rate curve and 79.19% for
prediction rate curve. Likewise, the FR model result shows that the AUC was 90.92% for success rate
curve and 84.38% for prediction rate curve. In the current study, both the models are having high
accuracy and both model are suitable for landslide susceptibility studies in the Hindu Kush region.

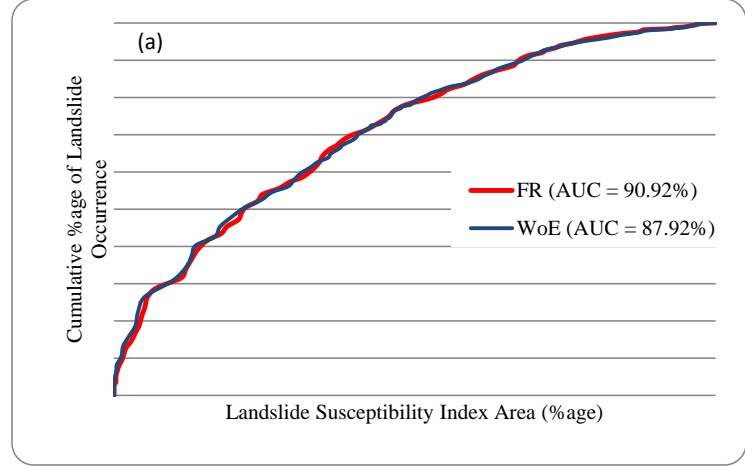


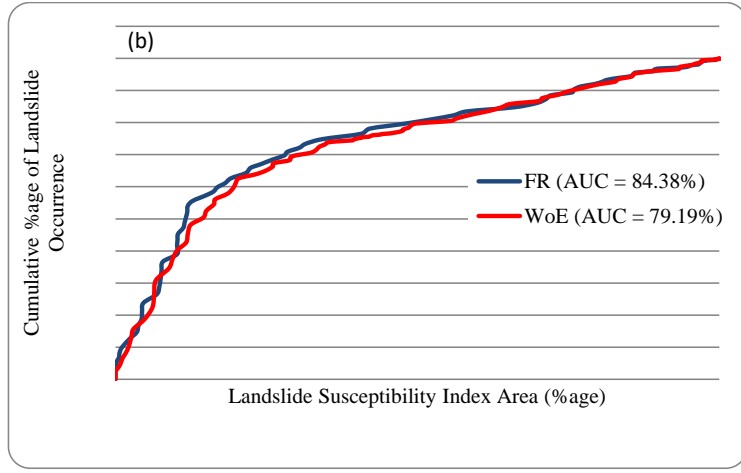


**Fig. 6.** Shahpur valley, (a) Success rate curve, (b) Prediction rate curve; showing the prediction
capability of WoE and FR models


## 5    Conclusion

In the current study frequency ratio and weight of evidence models were applied to develop
landslide susceptibility map. Initially, past landslides were identified from SPOT satellite image
and consecutive field visits and plotted on map. Landslide causative factors that were identified
from literature review including surface lithology, fault lines, land cover, slope gradient and
aspect, distance from streams and roads. The maps of these factors were prepared for
susceptibility analysis.  The roles of each class of these factor maps in landslide occurrence were
analyzed and assigned weights were calculated by implementing Bayesian probability models
i.e. weight of evidence and frequency ratio. The required susceptibility maps were generated
using $\sum W^c$ and $\sum FR$ values through overlay analysis in GIS.
The maps of landslide susceptibility were prepared based on both models and then validated
using success rate curve and prediction rate curve. It is further concluded that in Shahpur valley,
the results of frequency ratio model proved better than the weight of evidence model for
landslide susceptibility studies in the Hindu Kush region. This study can assist the disaster
management authorities to develop location specific mitigation measures for landslide hazards
to avoid loss of life and damages to infrastructure in future. The study conclude that landslide
hazard in the region may have negative impacts on agricultural activities, natural ecosystem, on
river morphology, human lives and infrastructure in the study area. In this regard proper land
use planning and strict forest preservation measures are highly required to reduce environmental
dama upges.

## Conflict of Interest

All authors have no conflict of interest.

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
