# Peer review of "Assessment of Landslide Susceptibility using Weight of Evidence and Frequency 1 Ratio Model in Shahpur Valley, Eastern Hindu Kush 2 1\*Ghani Rahman,2 Atta Ur Rahman, 3Alam Sher Bacha, 4Shakeel Mahmood, 3 5Muhammad Farhan Ul Moazzam, 5</su"

_Natural Hazards and Earth System Sciences, 2020_

## Referee Comment (RC1) · Anonymous Referee #1 · 23 Sep 2020

The paper NHESS-2020-167 deals with approaches to assess landslide susceptibility for an area of Pakistan. The methods are well-known, hence some novelty on the discussion and comments are required. In the following my comments:

1) You have to add the description of the geological setting in the paragraph of the study area. 2) It is very important that you describe what kind of landslides you are studying. You need to classify them (follow Hungr et al. or Varnes Classification). The type of landslide also affects the choice of the parameters to adopt for modeling. This will permit you to motivate the chosen parameters (see next point) 3) The adopted parameters require few words of motivation on their choice! 4) You wrote about field

survey, some figures of the studied events will definitely enrich the paper. 5) I did not understand the triggering of these landslides, earthquakes? In case of seismic-induced events probably a seismic hazard map could be added as parameter. 6) In the resulting map faults strongly control the results! I expect that they could have an important role if you deal with rockfalls and close to faults the rock mass is more fragmented, otherwise I can't understand their role very well. 7) What 's the criterion used to divide the susceptibility ranking? This is very important, several researchers worked on this topic.

Minor issues: line 46 pay attention to brackets line 113 almost every? what?

---

## Author Comment (AC1) · 10 Nov 2020

Respected Reviewer,

I have well received your valuable comments on our paper and we have revised it thoroughly. I just want to know that do we need to submit the revised paper now or we have to wait for the comments of the second reviewer.

Thanks.

---

## Referee Comment (RC2) · Anonymous Referee #2 · 19 Jan 2021

English has to be revised, many awkward sentences, a lot of repeated and/or obvious concept, use of non standard terminology or unusual words (e.g. "causative" instead of "causal"). Some suggestion for improvement added to the corrected text.

I don't know the meaning of the note (!!! INVALID CITATION !!!) it was put by another reviewer.

Not much new from the scientific point of view, actually the sole novelty is the study area.

to authors: form fig 5 it is obvious that faults are the most important factor for the landslides susceptibility in the study area, so some information and data on the

seismicity of the area would improve the paper value.

Please also note the supplement to this comment:
https://nhess.copernicus.org/preprints/nhess-2020-167/nhess-2020-167-RC2-supplement.pdf

──────────────────────────

[Figure]

**Supplement:**

Assessment of Landslide Susceptibility using Weight of Evidence and Frequency 1 Ratio Model in Shahpur Valley, Eastern Hindu Kush 2 1\*Ghani Rahman,2 Atta Ur Rahman, 3Alam Sher Bacha, 4Shakeel Mahmood, 3 5Muhammad Farhan Ul Moazzam, 5\*Byung Gul Lee 4 1Department of Geography, University of Gujrat, Pakistan 5 2Department of Geography, University of Peshawar, Pakistan 6 3National 
[revised manuscript text omitted]

English rewrite!

bad English

and not

clear

102

**Figure 1: Digital Elevation Map of Shahpur valley**

103 The uplift of HKH region came into existence due to the collision of Eurasian and Indian plate during the Cretaceous and Mio-Pliocene epoch. As a result of this collision these mountains are still 104 continuously rising at a rate of 4 to 5 mm/year (Jehan & Ahmad, 2006). There is high altitudinal 105 variation of 3600 meters in just 259 square km area (Figure 1). The valley has steep slope in the 106 upper part while it became gentle in the lower reach of the valley. The valley is drained by a stream 107 known as Khan Khwar. The study area consist of young mountain system that have immature 108 geology and is prone to landsliding phenomena which often results considerable property damages 109 and human losses almost every. The probability of these damages is expected to increase further as a laready 110 result of anthropogenic activities like deforestation, overgrazing, agricultural activities and 111 development of infrastructure in this area. Population growth has posed more pressure on the fragile 112 slopes and has made it more vulnerable for landsliding. 113

said

**3. Methods and Material**

In the eastern Hindu Kush region, Shahpur valley was selected for detailed analysis to grasp awkward 116 the governing landslide causative factors, which frequently trigger landsliding. The data from both 117 primary and secondary data sources were used to achieve the objectives of the study (Figure 2). The 118 past landslide sites were identified and mapped on 2.5m resolution SPOT image of April 2013. A 119 thorough field study was conducted to confirm the landslide sites on the ground and identify the 120 landslide triggering factors with local community knowledge. Seven triggering factors namely instability: 121 surface geology, proximity to fault line, slope gradient and aspect, land use/ land cover, nearness to 122 road and streams were identified. 123

Data regarding landslide triggering factors were acquired including the surface geology and 124 tectonics from geological map of North Pakistan. The administrative boundaries and settlement 125 126 shape-files was prepared from topographic sheets (RF 1:50,000) obtained from survey of Pakistan. Spatial features of roads network was acquired from the office of Communication and Works 127 128 Department, Peshawar. Land use/land cover map was obtained after applying supervised classification on SPOT satellite image using ArcGIS 10.2. ASTERGDEM having 30m was used for 129 extracting slope angle, slope aspect and hydrology of the study area. Furthermore, a detailed field 130 survey was conducted to validate the sites of already activated and potentially active landslide area. 131

GIS and Remote Sensing have been used for the preparation of spatial databases and landslides inventory map. Weight of evidence and frequency ratio model analysis is a bivariate statistical methodology in which the importance of each factor or combined factors is individually analyzed with respect to spatial distribution of existing landslides. The assumption in both models is that the factors which influenced the incidence of landslides in the past will be the same to trigger new landslides in future.

---

## Author Comment (AC2) · 16 Feb 2021

Reviewer # 1: The paper NHESS-2020-167 deals with approaches to assess landslide susceptibility for an area of Pakistan. The methods are well-known, hence some novelty on the discussion and comments are required. In the following my comments: 1) You have to add the description of the geological setting in the paragraph of the study area. Response: Thank you for your valuable suggestion. We have added a description of the geology of the area.

2) It is very important that you describe what kind of landslides you are studying. You need to classify them (follow Hungr et al. or Varnes Classification). The type of landslide also affects the choice of the parameters to adopt for modeling. This will permit you to motivate the chosen parameters. Response: Thank you for your suggestion we have classified the landslide.

3) The adopted parameters require few words of motivation on their choice! Response: Thank you for your comment and suggestion. The motivation to choose the particular parameters are added in the manuscript.

4) You wrote about the field survey; some figures of the studied events will definitely enrich the paper. Response: Thank you for your comment and suggestion we have added the field photographs in the appendix section of the manuscript.

5) I did not understand the triggering of these landslides, earthquakes? In the case of seismic-induced events probably a seismic hazard map could be added as a parameter. Response: Thank you for your valuable suggestion. We have added the seismic map in the manuscript.

6) In the resulting map faults strongly control the results! I expect that they could have an important role if you deal with rockfalls and close to faults the rock mass is more fragmented, otherwise I can't understand their role very well. Response: Thank you for your valuable comment. The seismic map is added in this regard (Figure 2).

7) What's the criterion used to divide the susceptibility ranking? This is very important, several researchers worked on this topic. Response: Thank you for making our attention towards it. We have used the natural break classification method to divide the landslide susceptibility index map. Minor issues: line 46 pay attention to brackets line 113 almost every? what? Reviewer # 2: 1) English has to be revised, many awkward sentences, a lot of repeated and/or obvious concepts, use of non-standard terminology or unusual words (e.g. "causative" instead of "causal"). Some suggestions for improvement added to the corrected text. Response: The English of the manuscript is extensively revised, all the awkward and the proper terminologies are added in the manuscript.

2) I don't know the meaning of the note (!!! INVALID CITATION !!!) it was put by another reviewer. Response: Thank you for your comment, actually some citations were missing, or maybe some problem with the endnote library. The citation problem is also fixed.

3) Not much new from the scientific point of view, actually the sole novelty in the study area. Response: Thank you for your comment. HKH region is geologically young mountains having high folds and faults as well as seismically active zone, therefore, a landslide is frequently occurring phenomenon in the study area

4) To authors: from fig 5 it is obvious that faults are the most important factor for the landslides susceptibility in the study area, so some information and data on the seismicity of the area would improve the paper value Response: Thank you for your valuable comment, the seismicity map of the region is added in the manuscript (Figure 2)

5) What was the pixel resolution for WoE and FR GIS models? Response: The pixel resolution for the selected methods are 30 meters.

6) Use a different name for the LSI from WoE and FR (e.g. LSI fr , LSIw ) Response: Thank you for your suggestion it is added in the manuscript.

Please also note the supplement to this comment:
https://nhess.copernicus.org/preprints/nhess-2020-167/nhess-2020-167-AC2-supplement.pdf

**Supplement:**

***The manuscript (nhess-2020-167), "Assessment of Landslide Susceptibility using Weight of Evidence and Frequency Ratio Model in Shahpur Valley, Eastern Hindu Kush"***

Reviewer # 1:

The paper NHESS-2020-167 deals with approaches to assess landslide susceptibility for an area of Pakistan. The methods are well-known, hence some novelty on the discussion and comments are required. In the following my comments:

1) You have to add the description of the geological setting in the paragraph of the study area.
Response: Thank you for your valuable suggestion. We have added the description about geology of the area.

2) It is very important that you describe what kind of landslides you are studying. You need to classify them (follow Hungr et al. or Varnes Classification). The type of landslide also affects the choice of the parameters to adopt for modelling. This will permit you to motivate the chosen parameters.
Response: Thank you for your suggestion we have classified the landslide.

3) The adopted parameters require few words of motivation on their choice!
Response: Thank you for your comment and suggestion. The motivation to choose the particular parameters are added in the manuscript.

4) You wrote about field survey; some figures of the studied events will definitely enrich the paper.
Response: Thank you for your comment and suggestion we have added the field photographs in the appendix section of the manuscript.

5) I did not understand the triggering of these landslides, earthquakes? In case of seismic induced events probably a seismic hazard map could be added as parameter.
Response: Thank you for your valuable suggestion. We have added the seismic map in the manuscript.

6) In the resulting map faults strongly control the results! I expect that they could have an important role if you deal with rock falls and close to faults the rock mass is more fragmented, otherwise I can't understand their role very well.
Response: Thank you for your valuable comment. Seismic map is added in this regard (Figure 2).

7) What 's the criterion used to divide the susceptibility ranking? This is very important, several researchers worked on this topic.
Response: Thank you for making our attention towards it. We have used natural break classification method to divide the landslide susceptibility index map.

**Minor issues**: line 46 pay attention to brackets line 113 almost every? what?

Reviewer # 2:

1) English has to be revised, many awkward sentences, a lot of repeated and/or obvious concept, use of non-standard terminology or unusual words (e.g. "causative" instead of "causal"). Some suggestion for improvement added to the corrected text.

   Response: The English of the manuscript is extensively revised, all the awkward and the proper terminologies are added in the manuscript.

2) I don't know the meaning of the note (!!! INVALID CITATION !!!) it was put by another reviewer.

   Response: Thank you for your comment, actually some citations were missing or may be some problem with the endnote library. The citation problem is also fixed.

3) Not much new from the scientific point of view, actually the sole novelty is the study area.

   Response: Thank you for your comment. HKH region is geologically young mountains having high folds and faults as well as seismically active zone therefore landslide is frequently occurring phenomenon in the study area

4) To authors: form fig 5 it is obvious that faults are the most important factor for the landslides susceptibility in the study area, so some information and data on the seismicity of the area would improve the paper value

   Response: Thank you for valuable comment, the seismicity map of the region is added in the manuscript (Figure 2)

5) What was the pixel resolution for WoE and FR GIS models?

   Response: The pixel resolution for the selected methods are 30 meters.

6) Use a different name for the LSI from WoE and FR (e.g. LSI fr , LSIw )

   Response: Thank you for your suggestion it is added in the manuscript.